# Gliadin Nanoparticles Containing Doxorubicin Hydrochloride: Characterization and Cytotoxicity

**DOI:** 10.3390/pharmaceutics15010180

**Published:** 2023-01-04

**Authors:** Silvia Voci, Agnese Gagliardi, Nicola Ambrosio, Maria Cristina Salvatici, Massimo Fresta, Donato Cosco

**Affiliations:** 1Department of Health Sciences, University “Magna Græcia” of Catanzaro, Campus Universitario “S Venuta”, 88100 Catanzaro, Italy; 2Institute of Chemistry of Organometallic Compounds (ICCOM)-Electron Microscopy Centre (Ce.M.E.), National Research Council (CNR), Via Madonna del Piano n. 10, Sesto Fiorentino, 50019 Florence, Italy

**Keywords:** cancer, doxorubicin hydrochloride, drug delivery, gliadin, nanoparticles, proteins

## Abstract

Doxorubicin hydrochloride (DOX) is a well-known antitumor drug used as first line treatment for many types of malignancies. Despite its clinical relevance, the administration of the compound is negatively affected by dose-dependent off-target toxicity phenomena. Nanotechnology has helped to overcome these important limitations by improving the therapeutic index of the bioactive and promoting the translation of novel nanomedicines into clinical practice. Herein, nanoparticles made up of wheat gliadin and stabilized by polyoxyethylene (2) oleyl ether were investigated for the first time as carriers of DOX. The encapsulation of the compound did not significantly affect the physico-chemical features of the gliadin nanoparticles (GNPs), which evidenced a mean diameter of ~180 nm, a polydispersity index < 0.2 and a negative surface charge. The nanosystems demonstrated great stability regarding temperature (25–50 °C) and were able to retain high amounts of drug, allowing its prolonged and sustained release for up to a week. In vitro viability assay performed against breast cancer cells demonstrated that the nanoencapsulation of DOX modulated the cytotoxicity of the bioactive as a function of the incubation time with respect to the free form of the drug. The results demonstrate the potential use of GNPs as carriers of hydrophilic antitumor compounds.

## 1. Introduction

DOX is the leading member of the anthracycline family which includes antibiotics characterized by a broad spectrum of anticancer activity [1]. It is a Food-and-Drug-Administration (FDA)-approved chemotherapeutic agent commercially available under the trade name of Adryamicin^®^, extensively used (alone or in combination with other compounds) for the treatment of a number of solid and liquid tumors [2]. Despite the advancements made in terms of new anticancer drugs discovered/synthesized, DOX still plays a leading role in cancer treatment. Indeed, it is included in the latest version of the World Health Organization list of essential medicines [3].

The cytotoxic activity of the drug is related to the arrest of the cell cycle thanks to the formation of a DNA-DOX adduct, and the inhibition of the activity of the Type II topoisomerase enzyme [4]. In addition, a reactive oxygen species-mediated mechanism (redox cycling) has been proposed. This is due to the conversion of the anthraquinone moiety of the drug into a semiquinone radical, which promotes an increased cell exposition to the radical species and leads to their death [5]. Unfortunately, since this activity is aspecific, severe adverse reactions against non-targeted tissues/organs have been observed. Indeed, dose-dependent cardiotoxicity and myelosuppression are the most common limiting effects of DOX administration [6,7]. Another important issue is related to the appearance of chemoresistant cell lines, a phenomenon caused by the upregulation of drug efflux transporters or epigenetic modifications [8,9].

In recent decades the entrapment of DOX in various types of drug delivery systems has been performed in order to bypass the aforesaid drawbacks, obtaining a controlled release of the active compound and a decrease of the administration times as well as an increased accumulation in tumor tissues [10]. To this end, liposomes, polymeric micelles, metallic- or solid-lipid nanoparticles have been developed, and some of them translated from bench to bedside. In this regard, Doxil^®^/Caelyx^®^ represents the milestone among the FDA and European-Medicinal-Agency (EMA)-approved nanomedicines [11,12,13].

The development of nanoformulations able to improve the pharmacological efficacy of antitumor drugs is a topic of great interest in pharmaceutical technology [14,15,16]. In this context, the selection of a biocompatible material to be used as the main component of the nanosystems plays a pivotal role, and the exploitation of natural polymers is continuously growing [17,18,19,20]. Among these, proteins are emerging as attractive biomaterials to be used to reach this goal [21]. This is mainly due to specific features of these macromolecules, such as their wide availability, great biocompatibility and opportunity to self-assemble into well-defined nanoaggregates by using simple and rapid preparation procedures and biocompatible solvents, while avoiding the use of chemical crosslinkers [22,23]. The last of these is of advantage because it avoids the toxicity phenomena associated with the presence of organic solvents and/or unreacted materials during the preparation procedure [22,23]. Moreover, protein nanoparticles favor the administration of lipophilic compounds in polar media and albumin-bound paclitaxel (Abraxane^®^) is probably the most noteworthy example [24].

Among the proteins, investigations concerning the use of wheat gliadin have increased [25]. This is due to the noteworthy emulsifying and mucoadhesive properties of the polymer, which, combined with its predominantly hydrophobic nature, have favored its exploitation in the biopharmaceutical and alimentary fields, especially for the delivery of water-insoluble compounds [26]. Nonetheless, to the best of our knowledge, only a very few investigations have focused on the encapsulation of anticancer drugs within gliadin-based nanoformulations such as all-*trans*-retinoic acid [27], cyclophosphamide [28], paclitaxel [29], diosmin [30,31,32] curcumin and methotrexate [33]. Indeed, hardly any active compounds have been tested.

For these reasons, the current study was designed to investigate the suitability of gliadin nanoparticles (GNPs) in antitumor therapy, and to this end DOX was used as a model of a water-soluble drug. Notwithstanding the inherent hydrophobic character of the polymer, our research group has recently demonstrated that polyoxyethylene (2) oleyl ether -stabilized gliadin nanoparticles can efficiently retain hydrophilic compounds [34,35]. The idea was to evaluate the physico-chemical and technological features of the DOX-loaded nanosystems in order to identify the best formulation to be used for in vitro experiments. And then the cytotoxic activity of the carriers was assessed against a model of breast cancer. The goal was to compare the antitumor features of the DOX-loaded nanoparticles with respect to those of the free drug, so as to be able to evaluate the benefits deriving from the nanoencapsulation of the compound in a gliadin-based matrix.

## 2. Materials and Methods

### 2.1. Materials

Gliadin from wheat, 3-(4,5-dimethylthiazol-2-yl)-3,5-diphenyltetrazolium bromide salt (used for MTT test), phosphate buffered saline tablets (PBS), dimethyl sulfoxide and amphotericin B solution (250 µg/mL) were all obtained from Sigma Aldrich (Milan, Italy). DOX was purchased from DBA Italia S.r.l. (Segrate, Milan, Italy) whereas Super Refined polyoxyethylene (2) oleyl ether was obtained from Croda International (Snaith, UK). Ethanol was purchased from Carlo Erba SpA (Milan, Italy). Dulbecco’s modified Eagle’s culture medium (DMEM) enriched with glutamax I, trypsin/ethylene diamine tetraacetic acid (EDTA), penicillin/streptomycin solution and fetal bovine serum (FBS), used during in vitro studies, were purchased from GIBCO (Life Technologies, Monza, Italy). Human breast adenocarcinoma cells (MCF-7) were obtained from the IRCCS Azienda Ospedaliera Universitaria San Martino—IST Istituto Nazionale per la Ricerca sul Cancro. Spectrum Laboratories Inc. (Eindhoven, The Netherlands) provided the cellulose acetate membrane used for in vitro release studies (cut-off 10 kDa).

### 2.2. Preparation of GNPs

GNPs were obtained by the nanoprecipitation technique, as previously described [34]. Briefly, gliadin (1.66 mg/mL) and polyoxyethylene (2) oleyl ether (0.1% *w/v*) were dissolved in 3 mL of a binary mixture of ethanol/water (2:1 *v/v*, pH 10). This solution was added to 5 mL of MilliQ water and then homogenized with an Ultraturrax (model T25 IKA^®^, Werke Gmbh & Co., Staufen, Germany) at 24,000 rpm for 2 min. The resulting suspension was mechanically stirred (8 h, 600 rpm, room temperature) with the aim of promoting the complete evaporation of the organic solvent. The final concentration of gliadin was 1 mg/mL.

DOX-loaded nanoparticles (DOX-GNPs) were obtained in the same way adding various amounts of the active compound (0.2–0.8 mg/mL) in the aqueous phase. The unencapsulated drug and/or the unreacted components were removed by means of ultracentrifugation (90,000× g for 1 h, 4 °C), using an Optima TL apparatus (Beckman Coulter s.r.l., Milan, Italy). The purification step was also performed before in vitro experiments using Amicon^®^ filters (cut-off 10 kDa, 4000 rpm for 40 min) [36].

### 2.3. Physico-Chemical Characterization

Photon Correlation Spectroscopy (PCS) was used to evaluate the mean diameter, size distribution and zeta potential of the systems using a 1:50 dilution and applying the third-order cumulant fitting correlation function (Zetasizer Nano ZS apparatus, Malvern Panalytical Ltd., Spectris plc, Malvern, UK) [37]. Transmission Electron Microscopy (TEM) was used in order to investigate the morphology of the nanosystems, as previously described [35,37].

Moreover, a Turbiscan Lab Expert^®^ apparatus (Formulaction, Toulouse, France) was employed with the aim of evaluating the stability of the various samples; the results were processed using Turby Soft 2.0 software (Formulaction, Toulouse, France) and expressed as Turbiscan Stability Index (TSI) as a function of time and temperature [35,38].

### 2.4. Entrapment Efficiency (EE), Loading Capacity (LC) and Release Profiles

The amount of DOX retained by GNPs was evaluated by UV-vis spectroscopy (Lambda 35, Perkin Elmer, Waltham, MA, USA). In detail, the calibration curve of the drug was obtained using various dilutions (140–5 µg/mL) of a stock solution of DOX (1 mg/mL in water). The calibration curve equation was
*y* = 0.0138*x* + 0.0808(1)
where *x* and *y* represent the known DOX concentration (µg/mL) and the relative absorbance, respectively. The obtained coefficient of linear regression (r^2^) was equal to 0.9983.

Each sample was centrifuged as described in Section 2.2, and the supernatant analyzed at a λ_max_ of 480 nm. The EE% was calculated as follows:EE (%) = D_e_/D_t_ × 100(2)
where D_e_ and D_t_ represent the amount of compound that became entrapped within the polymeric matrix and the amount of DOX initially added, respectively [35]. No interference between DOX and the other components of the GNPs was observed.

In addition, the iodine-iodide assay was applied in order to quantify the amount of Brij effectively integrated into the colloidal systems [23,35]. The drug loading capacity (LC) of the various formulations was calculated as the ratio percentage between the amount of entrapped DOX versus the total weight of the nanoparticles, according to the following equation:LC (%) = (Entrapped DOX)/(Total weight of nanoparticles) × 100(3)

The leakage of DOX from GNPs was evaluated by means of the dialysis method, at different pH and temperatures (pH 7.4, 37 °C and pH 5.5, 42 °C) [36,39]. In both cases, an isotonic solution of PBS 0.01 M was used, while the pH of the buffer was adjusted to 5.5 by the addition of hydrochloric acid 1 mol/L (pHmeter SevenCompact S210 Mettler Toledo).

Briefly, 1 mL of each formulation was placed in a dialysis bag and immersed into 200 mL of a constantly stirred and warmed PBS solution. The sink conditions were preserved. At fixed time points, 1 mL of the receptor fluid was withdrawn and replenished with an equal volume of fresh medium; the collected samples were then analyzed as described at the beginning of this section.

The amount of drug released was calculated as the ratio percentage between the amount of DOX found in the release media at each time point (DOX*_rel_*) with respect to the amount of drug entrapped within the particles (DOX*_load_*), as reported below:Release (%) = DOX*_rel_*/DOX*_load_* × 100(4)

In addition, the release kinetics of DOX from GNPs were investigated by fitting the results of release experiments using the Zero order, Higuchi, First order and Korsmeyer-Peppas mathematical models, as already described [40].

### 2.5. Cell Cultures and Cytotoxicity

MCF-7 cells were cultured in plastic culture dishes (100 mm × 20 mm) using a water-jacketed CO_2_ incubator (Thermo Scientific, Dreieich, Germany) at 37 °C (5% CO_2_) in a DMEM culture medium with glutamine, enriched with penicillin (100 UI/mL), streptomycin (100 µg/mL), amphotericin B (250 µg/mL) and FBS (10% *v/v*) [23].

The antitumor features of DOX (as an aqueous solution or encapsulated in GNPs), were assayed by MTT-testing. Namely, 7 × 10^3^ cells/well were seeded in a 96-multiwell culture plate and treated with various drug concentrations (0.01–5 µM) at different incubation times (24–72 h), as previously described [36,41].

Cell viability was evaluated by adding 20 μL of tetrazolium salts (solubilized in PBS at a concentration of 5 mg/mL) to each well, incubating the plate for 3 h at 37 °C and then analyzing the samples at 540 nm with a reference wavelength of 690 nm with a microplate spectrophotometer (xMARK^™^-BIORAD, Bio-Rad Laboratories Inc., Hercules, CA, USA). The cytotoxic profiles of the DOX-loaded GNPs were normalized with respect to those of the empty GNPs tested at the same concentrations.

The cell viability resulted from the mean of three different experiments ± standard deviation and expressed as follows:Cell viability (%) = (Abs*_T_*/Abs*_C_*) × 100(5)
in which Abs*_T_* is the absorbance of treated cells and Abs*_C_* is the absorbance of control (untreated cells).

Moreover, the concentration of DOX (in the free form or encapsulated within the carriers) that decreased the cell viability by half as compared to control (IC_50_), was calculated by fitting the MTT-test curves using the following four-parameter logistic equation (Sigmaplot 14.0, Systat software GmbH, Erkrath, Germany), as previously reported [42,43]
y = min + (max − min)/1 + (x/IC50)^−Hill slope^(6)
where:

y: is the response at the concentration x;

min: bottom of the curve;

max: top of the curve;

IC_50_: half-maximal inhibitory concentration;

Hill slope: is an absolute value that characterizes the slope of the curve at its midpoint.

### 2.6. Statistical Analysis

The statistical analysis of the various experiments was performed by ANOVA and the results were confirmed by a Bonferroni *t*-test, with a *p* value of <0.05 considered statistically significant.

## 3. Results and Discussion

### 3.1. Characterization of Nanosystems

GNPs prepared following the nanoprecipitation technique and stabilized by Brij O2, the non-ionic surfactant, have been recently proposed as potential delivery systems of various active compounds [34,35]. The main innovation of this formulation is based on the exploitation of a commercial-grade raw material, with the aim of obtaining wheat gluten-based nanoparticles characterized by a mean diameter of ~150 nm, a narrow size distribution (PdI ≈ 0.1) and a negative surface charge. The nanosystems entrapped large amounts of water-soluble compounds [34,35]; this last finding was contrary to what would be expected, considering the great hydrophobicity of the gliadin polymer which thereby can be dissolved only in binary ethanol/water mixtures [44].

Herein, GNPs were used for the encapsulation and delivery of DOX used as a model of a hydrophilic antitumor drug. In particular, the influence of the compound on the physico-chemical properties of the nanosystems was evaluated and reported in Figure 1A,B. The nanosystems evidenced no significant variations of the investigated physico-chemical parameters up to 0.6 mg/mL of drug. This data was supported by TEM analyses which evidenced a round-shaped morphology of the systems (Figure 1C,D).

Contrarily, greater amounts of the active compound exerted a negative influence on the particles promoting an increase of the mean diameter and polydispersity; indeed, a PdI of 0.3 was obtained when 0.8 mg/mL of compound was used during the preparation of the GNPs (Figure 1A).

The evaluation of the surface charge evidenced a progressive decrease of Zeta potential up to a DOX concentration of 0.6 mg/mL. The use of a greater amount of drug (0.8 mg/mL) promoted a slight variation of this parameter, probably as a consequence of the rearrangement of the colloidal structure. The fact that the samples were characterized by a negative surface charge could be advantageous for their potential application through intravenous administration (Figure 1B) [38,45].

The stability of the DOX-loaded GNPs was investigated as a function of time and temperature, by using the static multiple light scattering technique. The aim was to obtain qualitative information concerning the potential destabilizing phenomena occurring during the storage of the formulations [35,36]. Figure 2 shows the variation of the TSI slopes of each formulation, allowing an easy comparison of the behavior of the samples. In detail, no differences can be observed between the profiles of the empty and drug-loaded nanosystems when the analysis was performed at room temperature. A corroboration of this finding emerged when the transmission and backscattering profiles were evaluated (ΔT and ΔBS, respectively) (Appendix A). On the contrary, it was possible to observe that the sample prepared with 0.8 mg/mL of DOX evidenced a profile typical of an unstable formulation at 37 °C, since a consistent increase in its TSI slope occurred (Figure 2). This could be related to the adverse phenomena exerted by the heating process and by the substantial amounts of drug used during the sample preparation, as was also demonstrated by the reported ΔT and ΔBS profiles (Appendix A). The influence of temperature was further investigated using the PCS technique in the range between 30 and 50 °C (Table 1). The choice of this temperature range was due to the possibility that exists to use thermal ablation (50 °C) in cancer therapy to directly kill the malignant cells or to enhance the therapeutic response to chemo- or radiotherapy (adjuvant thermo-chemotherapy, 40–43 °C) [46,47,48]. In this context, it is necessary for the investigated formulations to show no or, at most, slight variations during the heating process in order to avoid any premature leakage of the entrapped drug, due to possible changes in the protein structure.

Figure 2 and Table 1 evidence the noteworthy stability of the nanoparticles which were characterized by a mean diameter of <200 nm and a narrow size distribution (0.1 ≥ PdI ≤ 0.2). In particular, it is possible to observe that the empty GNPs were not affected by the temperature. The samples prepared with an initial DOX amount up to 0.4 mg/mL followed the same trend, while the formulation obtained using 0.6 mg/mL of the active compound evidenced a slight increase in the sizes with respect to the empty systems. The comparison of the size distribution of the samples obtained at 37 °C and 50 °C showed overlapping trends; once again the nanoparticles prepared with 0.8 mg/mL of DOX demonstrated scarce physical stability (Figure 2). As previously observed, the surface charge of the various formulations did not evidence any variations upon heating (Table 1).

### 3.2. Retention Rate of DOX and Release Profiles

The second step of this investigation was based on the evaluation of the retention rate of DOX within the GNPs with the aim of selecting the most suitable formulation to be used for in vitro testing.

The data concerning the entrapment efficiency (EE) showed that when 0.2 mg/mL of DOX was used 60% of the drug became entrapped (~0.12 mg of drug/mg of nanoparticles), and this parameter reached almost 90% when the initial amount of compound was doubled (0.35 mg drug/mg of formulation). The highest drug retention was obtained when 0.6 mg/mL of DOX was initially added, because an encapsulation of 0.45 mg/mL of compound occurred; however, the same value (EE% ~74) was obtained when the drug concentration initially used was equal to 0.8 mg/mL, allowing an encapsulation of ~0.59 mg of the bioactive compound/mL of formulation (Figure 3); on the other hand, the physico-chemical characterization of the nanosystems demonstrated that this concentration promoted a significant increase of the average diameter, PdI and TSI values of GNPs (Figure 1A and Figure 2, respectively).

The LC of the samples confirm this trend and showed increasing values confirming the ability of the polymeric matrix to hold the drug. These profiles can result from non-covalent bonds between the protonated amine group of DOX and the negatively-charged carboxyl groups of the polymer, as already described for several other protein-based DOX-loaded nanoformulations [49,50,51,52,53].

Another plausible explanation was provided by Kayani and coworkers, who evidenced that following the desolvation of the β-lactoglobulin molecules, a reduction in the solubility of DOX takes place, and this promotes multiple drug-polymer interactions, favoring the entrapment of the compound within the polymeric matrix [54]. It is also reasonable to believe that the amino-sugar moiety of DOX (daunosamine) contributes to the retention of the compound on the part of the gliadin nanoparticles. Indeed, it has been recently demonstrated that hydrogen bonds between the aminoglycoside mycosamine and the threonine residues brought about by gliadin, favor suitable encapsulation profiles for the antifungal agent natamycin [55].

The release rate of the active compound from the GNPs was evaluated in buffers simulating both physiological and tumor environments, respectively, as previously reported [39,56,57].

As can be seen (Figure 4), the obtained profiles were quite similar and evidenced a biphasic drug leakage with a burst effect early on, followed by a sustained drug release over time. In particular, the higher the initial drug concentration in the formulation, the slower was the release obtained. The burst effect that took place is in agreement with previous reports on gliadin-based formulations [58,59] as well as with our previous findings [34,35], and can often be observed in polymeric drug delivery systems containing hydrophilic compounds [60].

It was shown that an acidic medium and the increase in temperature promoted a slight increase in the amount of DOX released in the initial phase of testing (10–20% at pH 5.5, 42 °C vs. 5–10% at physiological pH, 37 °C) (Figure 4); a small amount of DOX leakage was observed in both simulated conditions, thus suggesting that most of the drug remained confined in the polymeric matrix. 

Once again, this confirms the great stability of the carriers which are able to prevent any premature drug leakage into blood circulation, allowing the release of DOX only when the tumor area is reached.

In view of these results, the formulation prepared with an initial DOX concentration of 0.6 mg/mL was chosen as the ideal amount to obtain nanosystems to be used for further experiments. For this reason, the release profile of DOX from this system was analyzed by different mathematical models in order to obtain information concerning the mechanisms governing the leakage of the drug from the GNPs.

Based on the obtained R^2^, the model that best describes the DOX leakage from the investigated formulation was the Higuchi one, typical of matrix-based systems [61] (Appendix A). In fact, the amount of drug released over time was function of the square root of time and was governed by the diffusion of the active compound though the gliadin matrix in physiological as well as simulated tumor condition (Appendix A). This trend is in agreement with those described for other polymeric formulations proposed for the delivery of DOX [62,63,64,65].

### 3.3. In Vitro Cytotoxic Effects of DOX-Loaded GNPs

Considering the aforesaid results, the nanosystems prepared using an initial amount of drug of 0.6 mg/mL were used to perform the experiments of cytotoxicity. More in detail, the last phase of this study was aimed at comparing the antitumoral features of DOX-loaded GNPs with respect to those of the free drug.

Figure 5A shows that DOX brought about a significant decrease in cell viability (50–80%) after only 24 h when the drug concentration was >0.01 µM. Contrarily, the encapsulation of the active compound promoted a decrease of its cytotoxicity and similar effects with respect to the free drug were obtained only after 72 h incubation (Figure 5C). The evaluation of the IC_50_ confirmed this trend demonstrating a progressive enhancement of the efficacy of the nanosystems (Figure 5D). Similar results have been reported for other DOX-loaded nanosystems made up of bovine serum albumin [49], soybean protein isolate [50], hyaluronic [65] and poly-lactic-co-glycolic acid [66,67], polystyrene [68], hydroxyapatite [69], polyethylene glycol [70], as well as gliadin nanoparticles containing paclitaxel [29] and methotrexate [33].

In particular, the different cytotoxic profiles obtained with respect to the free form of the encapsulated molecules were explained as a function of the mechanisms that regulate the uptake of the free drug with respect to the various colloidal formulations. Specifically, DOX can enter cells by passive diffusion whereas nanoparticles are internalized through an energy-dependent process (endocytosis) [33,71,72]. Another plausible explanation is based on the capacity of GNPs to strongly retain their cargo molecule and release it in a slow and prolonged manner, as described in Section 3.2.

Moreover, the same trend has been reported for other antitumoral drug-loaded protein-based nanoformulations and will be of great advantage especially because of the possibility of decreasing both the necessary number of administrations and the potential side effects [23,33,49,56,66,73].

## 4. Conclusions

Reaching a high entrapment efficiency of a hydrophilic molecule in a colloidal system can be difficult to obtain because of the spontaneous tendency of the compound to partition into the aqueous phase of the formulation [74].

This work demonstrates that nanoparticles made up of commercial-grade gliadin can be used for the encapsulation of DOX. The proposed formulations evidenced characteristics suitable for their exploitation in anticancer therapy and were able to retain large amounts of drug while allowing its sustained release in media simulating physiological and tumor conditions, respectively. The strong interaction between the active compound and the protein promoted a decrease of the in vitro cytotoxicity of the drug. Although this last finding could be seen as detrimental towards supporting the nanoencapsulation of DOX in the proposed nanoparticles, especially when compared to other proposed formulations or those already approved for clinical application (Livatag^®^, Doxil^®^, Myocet^®^), the use of a natural, low-cost, raw material as well as the exploitation of a simple and scalable preparation procedure requiring the use of only bio-acceptable solvents, are the real advantages of this nanoformulation.

Additional studies concerning the in vivo behavior of the DOX-loaded GNPs are required in order to better evaluate their antitumor efficacy and investigate their pharmacokinetic profiles. Moreover, investigations aiming at the refinement of the surface characteristics of the nanoparticles by means of a stealth-coating layer are in progress in order to propose a gliadin-based formulation to be proposed as an innovative nanomedicine.

## Figures and Tables

**Figure 1 pharmaceutics-15-00180-f001:**
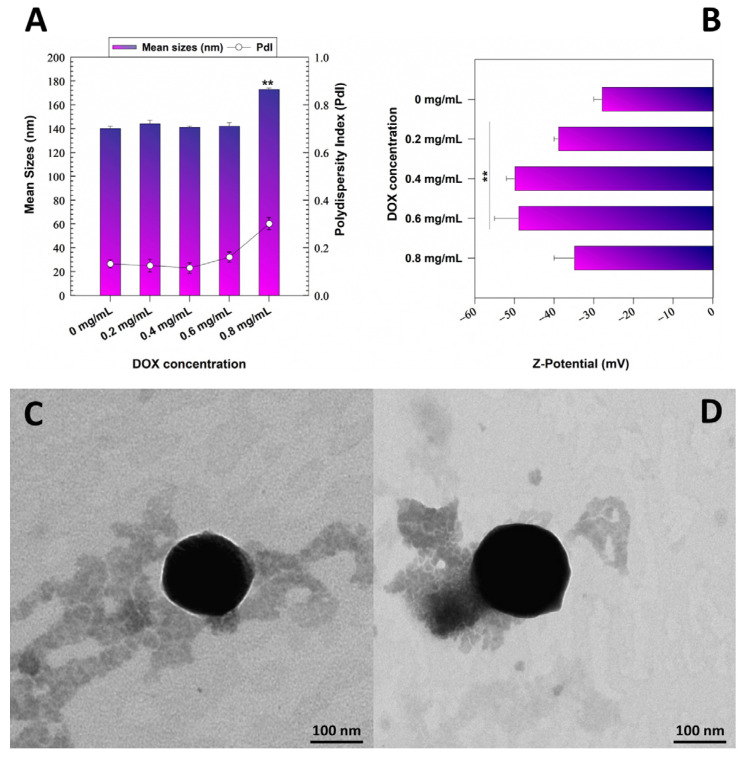
(**A**) Mean sizes, polydispersity index and (**B**) surface charge of GNPs (1 mg/mL of protein, 0.1% *w/v* of polyoxyethylene (2) oleyl ether) prepared with various amounts of DOX (0.2–0.8 mg/mL). Results are expressed as the mean of three independent measurements performed in triplicate on three different samples ± standard deviation. ** *p* < 0.001 with respect to the empty formulation. TEM images of (**C**) empty and (**D**) GNPs prepared with 0.6 mg/mL of drug.

**Figure 2 pharmaceutics-15-00180-f002:**
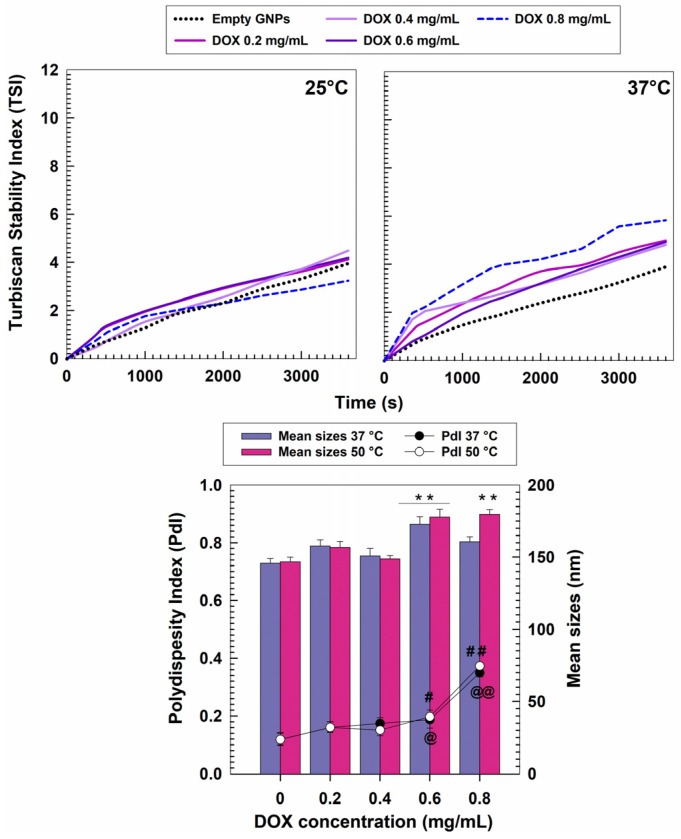
TSI profiles of gliadin nanoparticles (1 mg/mL of protein and 0.1% *w/v* of polyoxyethylene (2) oleyl ether) containing various amounts of DOX (0.2−0.8 mg/mL). The analyses were performed at room and body temperature for 1 h. Evaluation of the mean diameter and size distribution of gliadin nanoparticles as a function of the drug concentration and incubation temperature. Error bars if not shown are within symbols. Values are the result of three experiments ± standard deviation. ** *p* < 0.001 with respect to the mean sizes of the empty formulation analyzed at 37 °C and 50 °C, respectively. @ and # *p* < 0.05 with respect to the PdI of the empty formulation analyzed at 37 °C and 50 °C, respectively; @@ and ## *p* < 0.001 with respect to the PdI of the empty formulation analyzed at 37 °C and 50 °C, respectively.

**Figure 3 pharmaceutics-15-00180-f003:**
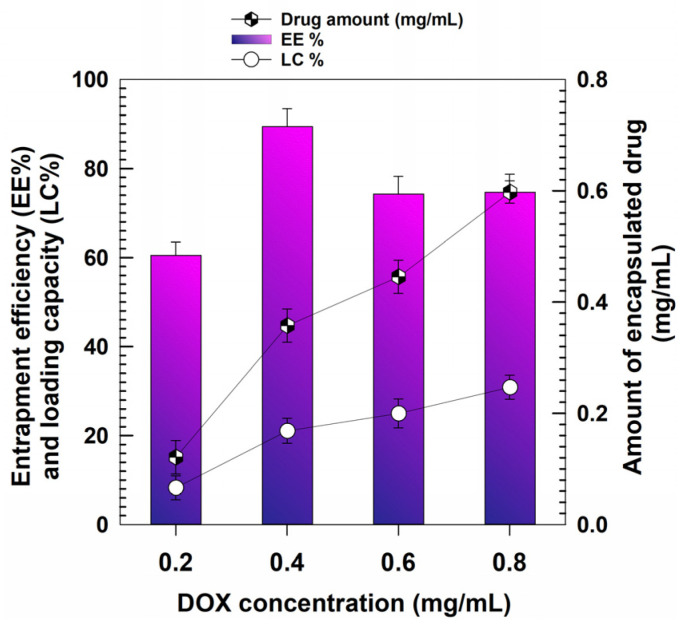
Entrapment efficiency and loading capacity of DOX as a function of the amount of drug initially used.

**Figure 4 pharmaceutics-15-00180-f004:**
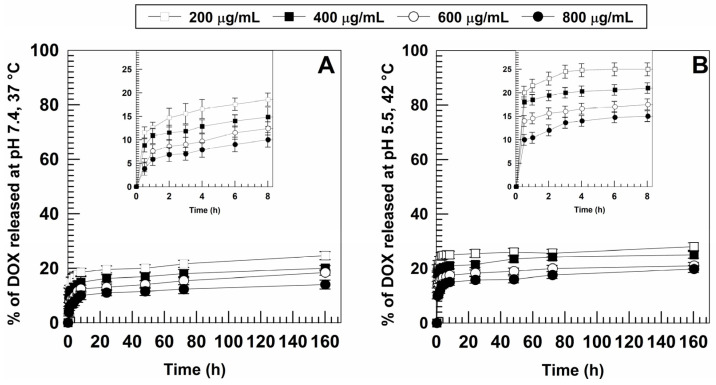
Release profiles of DOX (0.2–0.8 mg/mL) from gliadin nanoparticles (1 mg/mL of protein, 0.1% *w/v* of polyoxyethylene (2) oleyl ether) evaluated in (**A**) physiological and (**B**) in simulated tumor conditions and expressed as a function of the amount of DOX entrapped and incubation time. Error bars, if not shown, are within the symbols. Values represent the mean of five different experiments ± standard deviation.

**Figure 5 pharmaceutics-15-00180-f005:**
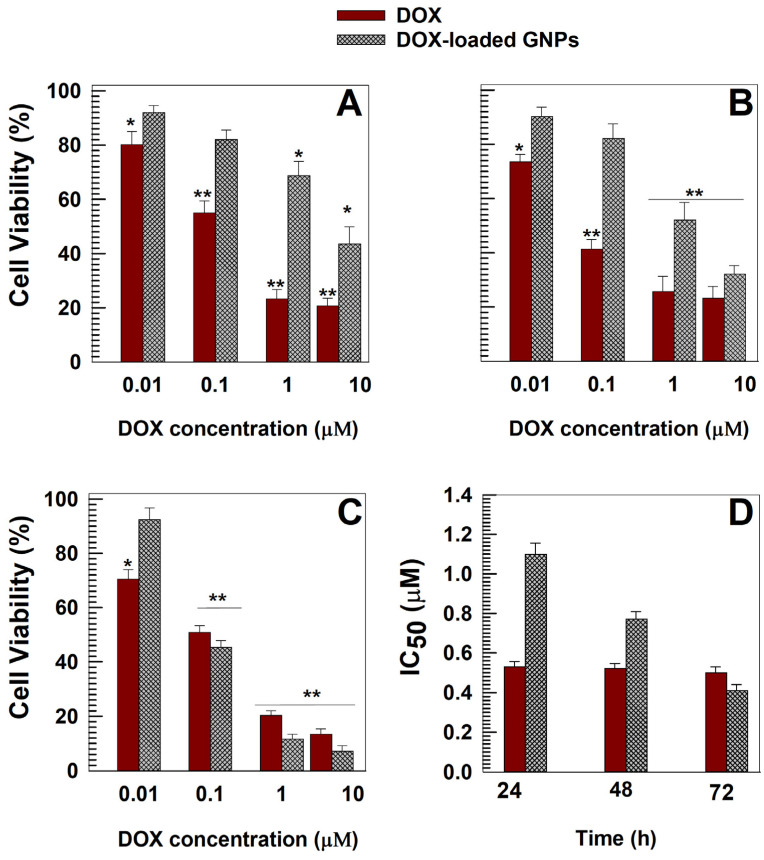
In vitro dose-dependent cytotoxicity of DOX and DOX-loaded GNPs evaluated by MTT testing on MCF-7 cells after (**A**) 24, (**B**) 48 and (**C**) 72 h of incubation. (**D**) Evaluation of the IC_50_ values (µM) of DOX and DOX-loaded GNPs Results are the mean of three different experiments ± standard deviation. * *p* < 0.05, ** *p* < 0.001 with respect to the untreated cells.

**Table 1 pharmaceutics-15-00180-t001:** Physico-chemical features of GNPs (1 mg/mL of polymer, 0.1% *w/v* of polyoxyethylene (2) oleyl ether) prepared with various amounts of DOX as a function of the incubation temperature.

DrugConcentration (mg/mL)	Temperature (°C)	Mean Sizes(nm)	PdI	Z-Potential(mV)
-	25	143 ± 4	0.131 ± 0.004	−28 ± 1
	30	149 ± 2	0.131 ± 0.040	−26 ± 1
	40	146 ± 3	0.120 ± 0.012	−25 ± 1
	50	147 ± 3	0.120 ± 0.020	−23 ± 1
0.2	25	144 ± 2	0.125 ± 0.027	−39 ± 1
	30	156 ± 2	0.124 ± 0.031	−49 ± 2 **
	40	158 ± 5 *	0.162 ± 0.017 *	−44 ± 1 *
	50	157 ± 4 *	0.162 ± 0.048 *	−45 ± 1 *
0.4	25	141 ± 1	0.115 ± 0.010	−50 ± 3
	30	145 ± 1	0.161 ± 0.021 *	−49 ± 2
	40	151 ± 5	0.176 ± 0.013 *	−49 ± 3
	50	149 ± 2	0.153 ± 0.042 *	−43 ± 1 *
0.6	25	142 ± 3	0.160 ± 0.021	−49 ± 2
	30	160 ± 3 *	0.197 ± 0.005 *	−42 ± 4 *
	40	173 ± 5 *	0.189 ± 0.031 *	−49 ± 2
	50	178 ± 2 *	0.199 ± 0.007 **	−44 ± 2
0.8	25	173 ± 1	0.301 ± 0.026	−35 ± 2
	30	158 ± 2 *	0.326 ± 0.004 *	−37 ± 4
	40	161 ± 3 *	0.351 ± 0.007 *	−39 ± 3
	50	180 ± 3	0.355 ± 0.040 **	−34 ± 1

The intergroup comparison of the various samples was performed by one-way ANOVA. * *p* < 0.05, ** *p* < 0.001 with respect to the formulation analyzed at 25 °C.

## Data Availability

All data available are reported in the article.

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
