# Peer review of "Gliadin Nanoparticles Containing Doxorubicin Hydrochloride: Characterization and Cytotoxicity"

_pharmaceutics, 2023, doi:10.3390/pharmaceutics15010180_

Round 1

Reviewer 1 Report

The manuscript describes the preparation, characterization, and cytotoxicity studies of nanoparticles of gliadin, a wheat protein, containing the antitumor drug doxorubicin.

The study is well structured and the tests carried out are suitable for this work. In addition, the authors present the results reliably and with pertinent discussions. Therefore I recommend the publication of this study in the journal Pharmaceutics after some minor consideration.

1- Why do authors use only the trade name (Brij O2) throughout the manuscript? It would be interesting to present the proper name of the compound.

2- In the in vitro release assays, the authors should incorporate a discussion about the transport mechanisms involved in the release of doxorubicin from nanoparticles. There are consolidated models in the literature that can be applied to the experimental data of this study and would add even more value to the work presented.

Reviewer 2 Report

The manuscript introduced how gliadin nanoparticles can be prepared as carriers of hydrophilic antitumor compounds (DOX). I have a few comments and questions:

I am curious about why the Mean Sizes of GNPs is much higher (than 0.6 mg/ml or lower) at DOX concentration = 0.8 mg/ml ? Can the author label "100 nm" on Figure 1 C and D directly?

In Figure 3, the author mentioned "The highest drug retention was obtained when 0.6 mg/mL of DOX was initially added, because an encapsulation (is that LC?) of 0.45 mg/mL of compound occurred" . We can see when 0.8 mg/mL of DOX was initially added, the LC is even higher, around 0.5 mg/mL. Can the author explain the details?

Can the author label (a) and (b) of Figure 4 and reorganize the caption of (a) and (b) ? The two different pH and temperatures (pH 7.4, 37° C and pH 5.5, 42 °C) were chosen from reference [36,39].  PH and temperature are both different in the two compared groups. Based on the authors' experiment, which condition is superior?

Can the author label (a)-(d) of Figure 5 and reorganize the caption from (a) to (d) ? It seems the cell viability is lower in DOX-loaded GNPs than DOX alone when DOX concentration is higher than 0.1 (μM) after 72 hours. What kind of risks can we foresee? 

(page 11) The author mentioned "supplementary Figure 3", but there is no Figure 3 in the supplementary materials.
